# Identification of Exogenous Nitric Oxide-Responsive miRNAs from Alfalfa (*Medicago sativa* L.) under Drought Stress by High-Throughput Sequencing

**DOI:** 10.3390/genes11010030

**Published:** 2019-12-26

**Authors:** Yaodong Zhao, Wenjing Ma, Xiaohong Wei, Yu Long, Ying Zhao, Meifei Su, Qiaojuan Luo

**Affiliations:** 1College of Life Science and Technology, Gansu Agricultural University, Lanzhou 730070, China; zhaoyaodong927@163.com (Y.Z.); wenjingma24@163.com (W.M.); zhaoying924@outlook.com (Y.Z.); sumeifei369@163.com (M.S.); 13919338538@163.com (Q.L.); 2Gansu Key Laboratory of Crop Genetic Improvement and Germplasm Innovation, Lanzhou 730070, China; 3Gansu Key Laboratory of Arid Habitat Crop Science, Lanzhou 730070, China; 4College of Business Administration, Kent State University, Kent, OH 44240, USA; ylong@kent.edu

**Keywords:** drought stress, alfalfa, exogenous nitric oxide, miRNAs, differential expression

## Abstract

Alfalfa (*Medicago sativa* L.) is a high quality leguminous forage. Drought stress is one of the main factors that restrict the development of the alfalfa industry. High-throughput sequencing was used to analyze the microRNA (miRNA) profiles of alfalfa plants treated with CK (normal water), PEG (polyethylene glycol-6000; drought stress), and PEG + SNP (sodium nitroprusside; nitric oxide (NO) sprayed externally under drought stress). We identified 90 known miRNAs belonging to 46 families and predicted 177 new miRNAs. Real-time quantitative fluorescent PCR (qRT-PCR) was used to validate high-throughput expression analysis data. A total of 32 (14 known miRNAs and 18 new miRNAs) and 55 (24 known miRNAs and 31 new miRNAs) differentially expressed miRNAs were identified in PEG and PEG + SNP samples. This suggested that exogenous NO can induce more new miRNAs. The differentially expressed miRNA maturation sequences in the two treatment groups were targeted by 86 and 157 potential target genes, separately. The function of target genes was annotated by gene ontology (GO) enrichment and kyoto encyclopedia of genes and genomes (KEGG) analysis. The expression profiles of nine selected miRNAs and their target genes verified that their expression patterns were opposite. This study has documented that analysis of miRNA under PEG and PEG + SNP conditions provides important insights into the improvement of drought resistance of alfalfa by exogenous NO at the molecular level. This has important scientific value and practical significance for the improvement of plant drought resistance by exogenous NO.

## 1. Introduction

As a perennial forage, alfalfa has the advantages of good nutritional quality and high yield [1]. It originated in Asia and Transcaucasia [2]. Alfalfa has been widely planted all over the world, and has now been identified as a leading forage crop [3]. As a legume plant, alfalfa can improve the soil planting environment by fixing atmospheric nitrogen, also reducing the concentration of diethylhexyl phthalate (DEHP) in the soil, thereby protecting the ecological environment [4]. Furthermore, the root system of alfalfa can effectively prevent water loss and soil erosion [5]. With the increasing demand for feed, alfalfa has been planted all over the country. However, as a global habitat problem, drought has seriously affected the yield and quality of alfalfa and restricted the development of the alfalfa industry [6]. Therefore, it is imperative to improve the drought resistance of alfalfa. Recently, the application of exogenous materials is one of the main strategies involved in this alleviation of drought stress [7].

Nitric oxide (NO) is an important bioactive molecule in plants, which is catalyzed and synthesized by nitric oxide synthase (NOS) and nitrate reductase (NR) [8]. NO is involved in plant germination, root and leaf development, respiration, morphological occurrence, and delayed aging [9]. Studies have shown that the addition of exogenous SNP (sodium nitroprusside) can increase the content of endogenous NO in plants, which improves the resistance of plants under salinity [10], drought [11], heavy metal [12], heat [13], and cold damage stress [14]. NO not only directly causes changes in protein and enzyme activity, but also regulates gene transcription and post-translational modification, thereby enhancing the ability of plants to recover from drought stress [15]. However, studies conducted on improving drought resistance of alfalfa by exogenous NO mainly focus on physiological and biochemical aspects [16,17], and have received less attention in molecule research, especially in terms of miRNA. The mechanism of exogenous NO enhancing the tolerance in alfalfa to drought stress has not been fully revealed.

miRNA is about 22 nt long, and acts on the target gene and negatively regulates its expression level [18]. The miRNAs synthesized by the classical and non-canonical pathways can be complementary to miRNA and can be used to degrade the target gene or inhibit the translation of the target gene by binding to RISC (RNA-induced silencing complex) [19,20]. With the increasing maturity and development of small RNA sequencing, more and more miRNA sequences have been discovered and included. As of March 2018, a total of 38,589 precursor sequences and 48,885 mature sequences of 271 species were included in the miRNA database, which will provide an important tool for the study of miRNA. In the existing reports, miRNAs appear in many essential biological metabolic processes in plants, such as seed germination and signal transduction [21,22]. The function of some miRNAs has also been identified. For example, overexpression of miR172 can seriously affect flower development [23], conserved miR159 controls leaf morphogenesis, and miR396 regulates abscisic acid (ABA) expression [24,25]. However, the functions of some miRNAs have not all been identified. There exists a need for further investigation. Many miRNAs arise when plants cope with various stresses. Research reporting on miRNA involvement in some plant species’ ability to cope with drought stress, such as tomato [26], potato [27], cotton [28], and *Medicago truncatula* [29], has been extensively studied. Due to the recent reports on the molecular mechanism of plant drought resistance, miRNA have garnered the interest of many experts and scholars.

Some drought-related miRNAs in leaves and roots of *Medicago sativa* ‘Aohan’ have been identified, and they have proven that miRNA plays an important role in regulating the drought resistance of alfalfa [30]. Muhammad Arshad et al. found that miR156 improved the drought resistance of alfalfa by acting on WD40-2 [31]. In addition, salinity-regulated [32], fall dormancy-related [33], freezing stress-related [34], and Pi deficiency-related [35] miRNAs have been identified in *Medicago sativa.* The above results suggest that miRNA plays a key role in the metabolism of alfalfa. This provides a theoretical basis and explains the exogenous nitric oxide-regulated miRNA expression in alfalfa drought resistance.

Because of exogenous NO function in the drought resistance of alfalfa and the potential value of miRNA in the stress resistance of alfalfa, it is boldly envisaged that exogenous NO regulates the expression of miRNA in alfalfa under drought stress. In our study, medium drought-tolerant variety *Medicago sativa* ‘San deli’ was used as the experimental material. Polyethylene glycol-6000 (PEG) was employed to simulate drought stress, along with spraying SNP (sodium nitroprusside, NO donor) externally. By analyzing the role of differentially expressed miRNA and their target sequence, the study aimed to (1) comprehensively screen the drought stress-responsive miRNA, (2) determine exogenous nitric oxide-regulated miRNAs under drought stress, and (3) make out the function of differentially expressed miRNA by analyzing the role of differentially expressed miRNA target genes. Exploring the mechanism of miRNA responding to NO under drought stress will provide a theoretical basis and technical support for the application of exogenous NO in plant stress resistance.

## 2. Materials and Methods 

### 2.1. Plant Materials and Growth Conditions

*Medicago sativa* L. ‘Sandeli’ was used in this study. Alfalfa plants were grown in pots with an a diameter of 15 cm, containing a mixture of nutrient soil and vermiculite (8:2 *v*/*v*), in a growth chamber at 25 ± 1 °C under a 12 h light and dark photoperiod. Plants were supplied with murashige-skoog (MS) nutrient solution. After alfalfa plants reached the age of 30 days, they were then randomly separated into three groups: CK (normal water), PEG (drought stress), and PEG + SNP (drought stress and SNP treatment) groups. Drought groups were conducted by watering 50 mL of 10% PEG solution once after every 2 days for 7 days. The PEG + SNP group sprayed 5 mL of 0.01 mmol/L SNP per day while drought was applied. Plants in the CK and PEG samples were simultaneously at 5 mL. Leaf samples from 30 plants in CK, PEG, and PEG + SNP groups were collected on the last day, and then immediately frozen in liquid nitrogen. Finally, they were then stored at −80 °C for further use. Three replicates from each experiment were used for the analysis.

### 2.2. Small RNA Library Construction and Sequencing

Total RNA was isolated from alfalfa leaves of the CK, PEG, and PEG + SNP-treated groups using Trizol reagent (Invitrogen, Carlsbad, CA, USA) according to the instructions. The integrity of the RNA samples was tested using an Agilent Bioanalyzer 2100 system (Agilent Technologies, SantaClara, CA, USA) to ensure sequencing with qualified samples. After the sample was tested, the amount of 2.5 ng was used as the starting amount of the RNA sample, the volume was supplemented to 6 μL with water, and the library was constructed using the Small RNA Sample Pre Kit. T4 RNA ligase 1 and T4 RNA ligase 2 (truncated) were ligated to the 3’ and 5’ ends of the small RNA, respectively. The complementary DNA (cDNA) was synthesized by reverse transcription, amplified by PCR, and the target fragment was screened by gel separation technique. The 18–30 nt fragment was a small RNA library. The concentration of the library was tested using Qubit 2.0, the library concentration was diluted to 1 ng/uL, and the insert size was detected using an Agilent 2100 bioanalyzer. The Q-PCR method was used to accurately quantify the effective concentration of the library to ensure library quality. After passing the library, the obtained RNA sequence was sequenced by Biomarket (Beijing, China) on the Illumina Hiseq 2500 platform, and the sequencing read length was single-end (SE) 50 nt. The raw data was saved in the National Center for Biotechnology Information (NCBI) sequence read file, and its Sequence Read Archive (SRA) data study deposit number is PRJNA551564.

### 2.3. Bioinformatic Identification of Conserved and Novel miRNAs

The original image data files obtained by sequencing were converted into Raw Reads by base call (Base Calling), and the results were stored in the FASTQ file format. High quality sequences (Clean Reads) were obtained by cleavage of the 3’ linker sequence and removal of sequences shorter than 18 or longer than 30 nucleotides. Using Bowtie (version 1.0.0) software, clean reads were sequenced with the Silva database (http://www.arb-silva.de/), the GtRNAdb database (http://lowelab.ucsc.edu/GtRNAdb/), the Rfam database (http://rfam.xfam.org/) and the Repbase database (http://www.girinst.org/repbase/); non-coding RNA(ncRNA) including ribosomal RNA (rRNA), transport RNA (tRNA), small nuclear RNA (snRNA), small nucleolar RNA (snoRNA) and repeated sequences were filtered to obtain unannotated reads containing miRNA; and mapped reads were obtained by sequence comparison with the transcriptome of alfalfa. In comparison with the mature miRNA sequences in known miRBase (version 21). reads that were identical to known miRNAs were considered as identification of known miRNAs. Potential new miRNAs have been obtained by predicting the remaining readings by the online prediction software miRDeep2 (https://github.com/hangelwen/miR-PREFeR).

### 2.4. Differential Expression Analysis of Known and Novel miRNAs

Differential expression analysis of mature miRNAs between CK and PEG samples, and CK and PEG + SNP samples was performed by DESeq (version 1.18.0) software. Differentially-expressed miRNAs were calculated using the expression of transcript per million (TPM) value, the expression of miRNA in treatment and control was statistically analyzed, and the expression was normalized by TPM algorithm. TPM values were calculated as follows: (actual miRNA count/total count of mapped reads) × 1,000,000. Then, the fold-change and *p*-values were calculated from the normalized expression, and the log2-ratio and scatter plots were then generated. A miRNA whose expression fold change satisfies | log2 (fold change) | ≥ 1 and whose *p*-value < 0.05 adjusted by the Benjamini–Hochberg method is considered as a differentially expressed miRNA.

### 2.5. Prediction of miRNA Targets, Gene Ontology (GO), and KEGG Pathway Analysis

The plant miRNA target gene prediction software TargetFinder (version 1.6) was used to predict the target genes of differentially expressed miRNA. Mature miRNA sequences were used as queries to search for potential target mRNAs in the Medicago sativa transcriptome, and default parameters were used for target prediction. By using BLAST (version 2.2.26) sequence alignment software, the predicted target gene sequence was compared with the database for NCBI non-redundant protein sequences (NR) (ftp://ftp.ncbi.nih.gov/blast/db/), swiss-prot (http://www.uniprot.org/), GO (http://www.geneontology.org/), clusters of orthologous groups (COG) (http://www.ncbi.nlm.nih.gov/COG/), KEGG (http://www.genome.jp/kegg/), protein homologous cluster (KOG) (http://www.ncbi.nlm.nih.gov/KOG/), and homologous protein family (Pfam) (http://pfam.xfam.org/) to obtain the annotation information of the target gene.

### 2.6. Validation of miRNA and Target Gene Expression with qRT-PCR

The expression of some miRNAs and target genes were validated by quantitative real-time PCR (qRT-PCR). qRT-PCR was performed on Analytikjena -qTOWER2.2 fluorescence quantitative PCR instrument (Germany). Similar RNA samples were used with Illumina sequencing in qRT-PCR analysis. Reverse transcriptase reactions contained DNase-treated total RNA and a gene-specific stem-loop real-time (RT) PCR primer. According to the instructions, reverse transcription was performed using a TUREscript 1st Stand cDNA SYNTHESIS (Aidlab biotechnologies CO. Ltd, Beijing, China) kit using a 20 μL reaction system: total RNA 1000 ng, 4 μL 5 × RT Reaction Mix, 0.8 μL Rondam primer/oligodT, 0.8 μL TUREscript H- RTase/RI Mix, and finally adding RNase-Free dH_2_O to 20 μL. Reverse transcription was performed at 42 °C for 40 min, then reverse transcription at 65 °C for 10 min to obtain cDNA. The following temperature program was used to perform the RT reaction for 3 min at 95 °C, 40 cycles of 10 s at 95 °C, 30 s at 58 °C, and then the holding time was 4 s at 65 °C. All reactions were repeated three times. The mtr-miR7701-5p and c115758.graph_c0 were served as miRNA and mRNA for internal control, respectively. The relative expression of the target gene in each sample was calculated automatically by the instrument software qPCRsoft3.2, and manually calculated. The relative expression level of miRNAs and the predicted target genes was calculated using the 2 ^−ΔΔCT^ method, and relative expressions were shown as log_2_ fold change. The miRNAs and target genes primers are listed in Appendix A.

## 3. Results

### 3.1. Deep Sequencing Results of Alfalfa Small RNA (sRNA) Libraries

A total of 9 sRNA libraries (one library with weak correlation in CK and PEG + SNP treatment groups was deleted to ensure the accuracy of the subsequent analysis) comprising three samples (CK: leaves with normal water, PEG: leaves with drought stress, PEG + SNP: leaves with drought stress and SNP treatment) were generated using the Illumina HiSeq 2500 platform, and the sequencing data of the three samples were averaged. By extracting both the adapter and low amount of reads, the three samples obtained 23,528,741 (accounting for 87.63% of raw reads), 18,046,004 (accounting for 74.41% of raw reads), and 15,076,692 (accounting for 73.49% of raw reads) clean reads. High-quality sRNA reads were mapped to the alfalfa transcriptome sequence using Bowtie software. The CK, PEG, and PEG + SNP libraries were matched to 10,366,134, 7,651,367, and 6,487,853 mapped raw. In order to classify the sRNA sequenced reads into different categories and identify all the miRNA sequences in the libraries, the researchers mapped the reads to specific databases with Bowtie. At all levels, the tRNA, miRNA, snoRNA, and rRNA reads were annotated as the clean sRNA in these libraries. The reads of 1,267,908, 1,180,088, and 871,935 miRNAs were mapped to the CK, PEG, and PEG + SNP groups, respectively. Therefore, the percentage obtained for miRNA reads in PEG were more abundant than CK and PEG + SNP groups. Table 1 shows the details of raw reads, clean reads, and sRNA sequencing in three samples.

sRNA of different lengths have different functions. For example, 21/22 nt sRNAs are involved in post-transcriptional gene silencing and mRNA cleavage, whereas 24 nt sRNAs are primarily responsible for transcriptional gene silencing and DNA methylation. As seen in the distribution of sRNA length, 21 to 24 nt accounted for the majority, and the 24 nt category was the most abundant compared to other lengths, followed by 21 nt and 22 nt (Figure 1). As shown in Figure 1, the lengths of sRNA from CK, PEG, and PEG + SNP showed similar rules, but some slight differences still exist. The order of 24 nt sequences in each sample was found to be CK > PEG + SNP > PEG, and the 21–22 nt sequence exhibited an opposite expression profile compared to 24 nt. These consequences indicated increased levels. When alfalfa was subjected to drought stress, 21, 22, and 24 nt miRNAs reduced their role in post-transcriptional gene silencing, whereas exogenous nitric oxide can raise 24 nt sRNA-mediated transcriptional gene silencing, causing RNA-directed DNA methylation to occur.

### 3.2. Conserved miRNAs from Alfalfa

By comparing the known plant miRNAs in miRBase, the study identified some conserved miRNAs obtained by high-throughput sequencing. A total of 90 unique mature miRNA sequences from 46 families were found in alfalfa. We found 88, 86, and 82 miRNAs from three treatment groups, and they belonged to 44 miRNA families. The miR530 family specifically expressed in the CK and PEG + SNP were guided by the miR2632 family, as shown in Appendix A. In comparison with other miRNA family readings, miR2109 and miR159 showed the highest level (Appendix A). The miR156 family contained 8 miRNAs with the most members. The abundance of conserved miRNAs of the top 10 most highly expressed in all libraries is shown in Figure 2.

### 3.3. Check of Unknown miRNAs in Medicago sativa L. “Sandeli”

To evaluate unknown miRNAs in the clean reads, miRDeep2 was used to forecast putative new miRNAs after removing the verified unknown miRNAs. The total of 177 new miRNAs were discovered. In the library of CK (169), PEG (172), and PEG + SNP (154), novel miRNAs were detected as well. Detailed information about the unknown miRNA sequences, extent of reads, pre-miRNA sequences, genome ID, miRNA length, and Fragments Per Kilobase Million (FPKM) are listed in Appendix A. Most new miRNAs have lower readings when compared to conserved miRNAs, and only 9 novel miRNAs had more than 1000 reads (Appendix A). Table 2 shows information of 10 new miRNAs, including length, score total, expressional reads, guanine (G )+ cytosine (C) contents, and hairpin energy, with high confidence found in all libraries.

### 3.4. The Response of miRNA to Drought Stress and Exogenous Nitric Oxide under Drought Stress

To identify miRNAs with response to drought and exogenous NO response under drought stress, the study compared the expressions of known and new miRNAs in CK and PEG, and CK and PEG + SNP samples. The researchers detected 32 miRNAs that responded to drought treatment (including 14 known miRNAs and 18 newly predicted miRNAs) and 55 miRNAs that responded to exogenous NO under drought stress (including 24 known, 31 unknown). Under PEG treatment, 11 of the 14 known miRNAs were up-regulated and 3 were down-regulated (Figure 3a, Appendix A); from the total of 18 differentially expressed new miRNAs, 8 were up-regulated and 10 were down-regulated. Under the treatment of PEG + SNP, 14 of the 24 known miRNAs were up-regulated and 10 were down-regulated (Figure 3b, Appendix A); 14 of the 31 new miRNAs were up-regulated and 17 were down-regulated. Detailed information of the drought-responsive and exogenous nitric oxide-responsive responses under drought stress miRNAs, such as FPKM, log fold-change, and *p*-value, are shown in Appendix A, respectively.

From the Appendix A, a huge number of miRNAs were not only silenced, but were induced by the NO under drought stress. The researchers identified 10 conserved miRNAs that were only regulated by exogenous NO under drought stress (Table 3). Four of them were silenced, including mtr-miR398a-5p, mtr-miR5232, mtr-miR5295a, and mtr-miR5559-5p. In addition, mtr-miR156a, mtr-miR399a, mtr-miR399c, mtr-miR399q, mtr-miR5213-5p, mtr-miR5752a, and mtr-miR7696a-5p were induced. More members of the miR399 family were regulated by exogenous NO. It must be mentioned that the expression of the homologous miRNAs, belonging to the same miRNA family, was consistently similar under drought stress. For instance, the miR156 family was all up-regulated, whereas the miR399 family was all down-regulated.

In addition, Figure 4a shows that 10 differentially-expressed conservative miRNAs were shared by PEG and PEG + SNP, but their FPKM were different. This indicated that the miRNAs were prerequisite in responding to drought stress, such as mtr-miR1510a-5p, mtr-miR172a, and mtr-miR408-3p. Figure 4b shows that common 14 miRNAs (7 raised and 7 cutted) were discovered in PEG and PEG + SNP samples. Novel miRNAs in PEG + SNP were more than those in PEG. This indicated that NO caused the emergence of more novel miRNAs. Overall, these new miRNAs may aid in clarifying the drought-responsive and exogenous nitric oxide-regulative mechanism under drought stress.

### 3.5. Prediction and Functional Classification of Target Genes of Drought Stress Response Type and Exogenous Nitric Oxide Reactive miRNA under Drought Stress

The current study used the TargetFinder server to predict the target sequence of differentially expressed miRNAs. A total of 86 and 157 putative targets were retrieved for differentially-expressed miRNAs in PEG and PEG + SNP, respectively. Appendix A show miRNA and corresponding target genes. The results reported that one miRNA can act on multiple mRNAs, and different miRNAs can target the same target gene (Appendix A). Due to lack of genome information, some miRNA targets were not predicted.

For example, Appendix A shows that squamosa promoter-binding-like protein (SBP) transcription factors were targeted by mtr-miR156a\e\g-5p in the PEG + SNP sample, and was involved during plant growth, morphogenesis, stress response, floral organ development, and plant photoreaction. However, only mtr-miR156e\g-5p regulates SBP transcription factors under drought stress [45,46]. This indicates that NO can induce more miRNA and its target genes to participate in drought resistance. In addition, exogenous NO specifically regulated myb-like DNA-binding domain (MYB) transcription factors under drought stress, which has shown various cellular processes involved in regulation including cell cycle, cell morphogenesis, and biotic and abiotic stress responses [47,48]. In particular, MYB transcription factors play an important role in phenylpropanoid metabolism in plants [49]. The results predicted that target genes are also involved in alfalfa phenylpropanoid metabolism.

To further comprehend the potential effect of drought-responsive and exogenous nitric oxide-regulative miRNAs in alfalfa, GO and KEGG pathway analyses were carried out on all target genes. The results indicated that 59 genes have been annotated and 114 GO terms were enriched under drought stress (Appendix A), including 68 biological action processes, 33 molecular functions, and 13 cell compositions. Most enriched GO terms were related to plant stress tolerance, for instance, defense response (GO:0006952), response to water deprivation (GO:0009414), respiratory burst involved in defense response (GO:0002679), and carbohydrate transport (GO:0008643). GO terms related to hormone metabolism were also enriched, including response to brassinosteroid (GO:0009741), abscisic acid-activated signaling pathway (GO:0009738), ethylene-activated signaling pathway (GO:0009873), and ethylene biosynthetic process (GO:0009693).

Many GO terms were also differently enriched in PEG + SNP samples. A total of 114 genes were annotated and 214 GO terms were regulated by extraneous nitric oxide under drought stress. Also, 117 biological reaction processes, 70 molecular functions, and 27 cell compositions were enriched (Appendix A). From Figure 5, in PEG + SNP samples, the number of biological processes, molecular functions, and cellular components increased by 69.57%, 1.33 and 1.08 times more than PEG samples, respectively, and exogenous NO regulated the initiation of more genes and multiple biological processes to respond to drought stress. Participation of more cellular components will aid alfalfa in improving the efficiency of drought response. As compared with drought stress, exogenous NO regulated carbohydrate metabolism in particular, which included the cellulose biosynthetic process (GO:0030244), starch biosynthetic process (GO:0019252), galactolipid biosynthetic process (GO:0019375), amylopectin biosynthetic process (GO:0010021), and carbohydrate transmembrane transport (GO:0034219).

A total of 6 and 12 KEGG pathways for the miRNA target gene were enriched in PEG and PEG + SNP samples (Appendix A), respectively. It must be noted that oxidative phosphorylation, phenylpropane metabolism, ubiquinone and other terpenoid metabolism, alpha-linolenic acid metabolism, and RNA transport were all enriched in both samples. Endocytosis was specifically enriched only in PEG samples. In PEG + SNP samples, exogenous NO specifically regulated metabolic pathways such as ascorbic acid, tyrosine metabolism, starch and sucrose metabolism, amino acid synthesis, ribosome, and protein transport in response to drought stress.

### 3.6. Validation of the Expression of miRNAs and Their Targets

In order to verify the expression of miRNA and its target genes, 9 miRNAs (mtr-miR5214-3p, mtr-miR2199, mtr-miR396a-5p, mtr-miR399b, mtr-miR408-5p, mtr-miR5752a, unconservative_c105676.graph_c0_4408, unconservative_c103008.graph_c0_2076, and unconservative_c103018.graph_c0_2090) and their target sequences were chosen for qRT-PCR validation (Figure 6). Except for mtr-miR5214-3p, the other qRT-PCR results of miRNA profiles were similar to the sequencing results. The target gene was expressed in the opposite way. For example, deep sequencing results exhibited that mtr-miR2199 was down-regulated and their corresponding target gene c114003.graph_c0 showed raising patterns. mtr-miR408-5p and mtr-miR5752a were significantly up-regulated, and for the corresponding target gene of interest c115697.graph_c0 and c117794.graph_c0 showed decreasing patterns. The results of qRT-PCR confirmed that the RNA sequencing results were authentic.

## 4. Discussion

Drought stress is one of the environmental stresses hindering the growth and development of plants [50]. There are three types of interactive modification that respond to drought stress in plants: the alteration in gene expression (over, under, and co-expression), dynamics in protein metabolism, and alterations in the metabolic pool [51,52]. The three types of modification above may all confer plant resistance or tolerance to drought stress. Studies have confirmed that miRNA can regulate gene expression, protein metabolism, and metabolic pool. In addition, sequencing techniques use for small RNAs improve the abundance of miRNA species. Many researchers have begun to utilize miRNA to reveal the drought-resistance mechanism of plants, such as *Arabidopsis* and *Oryza sativa* [38,53]. Functioning as second messenger of plants, NO release SNP agent, which has been shown to improve plant stress resistance. However, the report for miRNAs did reveal whether the molecular mechanism of exogenous NO can improve alfalfa drought resistance.

The current study showed that nine libraries were constructed from alfalfa treated with normal watering, drought stress, and exogenous NO under drought stress, and also that two uncorrelated libraries were discarded. All the libraries were sequenced on the Illumina Hiseq 2500 platform. The researchers identified that the known and new miRNAs induced by exogenous NO were under drought stress and drought correlation, and their predicted target genes were classified accordingly. According to the sequence, approximately 15–23 million clean reads for each library were generated. The reads for drought stress samples were decreased and the control, known miRNAs reads containing the same tendency—indicated that the drought treatment restrained expression of some miRNAs. Therefore, more miRNA-related genes and pathways responded to drought stress, which was similar to several previous studies on *Medicago sativa* L. cv. Aohan and tomato [26,34]. Coincidentally, exogenous NO enhanced this trend to decrease the harming of drought stress.

The results reported that 88, 86, and 82 miRNAs were obtained from the CK, PEG, and PEG + SNP groups in alfalfa, respectively. Among them, the miR156 family reported the highest members. The study also found that mtr-miR5213-5p, mtr-miR159a, mtr-miR398b, mtr-miR396a-5p, mtr-miR167b-5p, mtr-miR2643a, mtr-miR166a, mtr-miR7696a-3p, mtr-miR7696c-3p, and mtr-miR396b-5p were the most enriched miRNAs in response to drought stress. These highly expressed miRNAs may regulate gene expression. It is worth mentioning that mtr-miR5213-5p were the highly expressed miRNAs—this is consistent with Li et al.’s study in alfalfa under natural drought stress [30]. Previous studies have shown that mtr-miR5213-5p also regulated gene expression of alfalfa in response to salt-alkali stress [40]. These indicate that the expression of mtr-miR5213-5p is important in responding to drought and saline-alkali stresses.

Furthermore, the researchers predicted 177 new miRNAs in all libraries, and the expression levels of most new miRNAs were lower, which is consistent with previous reports [54,55]. Previous results are limited in terms of analyzing the differential expression of new miRNAs between different libraries. The possible reason is that novel miRNAs can only be induced to express under certain conditions such as environmental factor induction, and it is suggested that further investigation is needed to explore the expression mechanism.

The researchers compared the expression profiles of mature miRNAs in PEG and PEG + SNP samples with the control group (CK) to identify differentially expressed miRNAs. Differing from the expression of miRNAs under natural drought, the present results reported that 14 known miRNAs in the PEG sample and 24 known miRNAs in the PEG + SNP sample were considered as drought-responsive and nitric oxide-responsive under drought stress miRNAs [30]. The results also reported that two known miRNAs were identified by PEG-simulated drought treatment as compared to natural drought. This may have been due to the different manners of drought treatment. In addition, it was confirmed that miR408 and miR2118 in alfalfa leaves are associated with drought, and specifically identified in mtr-miR172a, mtr-miR398a-3p, mtr-miR5295a, and mtr-miR398a-5p, which are the drought-responsive miRNAs. Similar results were reported for the tomatoes, *Lycopersicon esculentum*, *Medicago sativa,* and *Medicago truncatula* [32,56]. Possible causes leading to these different consequences are (a) different alfalfa varieties, (b) different treatment conditions (natural drought versus PEG treatment), (c) different alfalfa developmental stages, (d) different growth environment, and (e) possible false positives in sequencing. In addition, miR1510 and miR5559 are specifically found in leguminous plants [30], as the current results showed that mtr-miR1510a-5p and mtr-miR5559-5p were identified and differentially expressed under drought stress. The above results are conducted at the level of log_2_ fold change > 1, *p*-value < 0.05.

Exogenous NO can induce mtr-miR156a, mtr-miR399a, mtr-miR399c, mtr-miR399q, and mtr-miR5213-5p down-regulated expression in alfalfa under drought stress. Past studies showed that the inhibition of gene translation or degradation of targeted mRNA can achieve miRNA regulation of gene expression [20]. It was predicted that the miRNA targets whose exogenous NO is down-regulated may play a positive regulatory role in drought stress response. mi156a are important miRNA for plant for abiotic stress responses, growth, and development [57]. miR156 can regulate plant stress tolerance by modulating SQUAMOSA promoter-binding-like protein (SPL) genes that function in anthocyanin biosynthesis [36]. We thought exogenous NO induced lower expression of miR156, which may positively regulate anthocyanin biosynthesis by raising the expression of SPL transcription factor. This is consistent with the results of our previous physiological experiments, and exogenous nitric oxide indeed promoted the synthesis of anthocyanins in alfalfa leaves in response to drought stress.

Exogenous NO may decrease due to the expression of miR399 in alfalfa to enhance drought tolerance. Plants suffering from water-deficiency usually show reduced uptake of mineral nutrients, and miR399 is involved in the regulation of phosphate (P) homeostasis [58]. It has been described that miR399 has a negative regulation with inorganic phosphate concentration by targeting a pyrophosphatase. Most researches have shown that AVP1 (*Arabidopsis* vacuolar pyrophosphatase gene) is overexpressed in a variety of plants, with enhanced drought and salt tolerance, such as *Arabidopsis* and rice [59,60]. Consistent with previous findings in sugarcane [61], the current study results showed that the expression of miR399 family was decreased in the alfalfa under drought stress. Exogenous nitric oxide, which inhibits the expression of mtr-mir399a /c/q, can also increase pyrophosphatase levels and promote the proton pump activity. As it promotes proton pump activity, the water potential in the plant vacuole can lower the activity of secondary transporters. In addition, this can promote the flow of ions in the cytoplasm [61]. The present results reported changes in the transporter activity.

The researchers of the present study speculated that exogenous NO may regulate the expression of miRNA398 to eliminate ROS produced by drought stress. mtr-miR398a-5p is an important member of the miR398 family. The miR398 family in *Medicago sativa* was up-regulated under drought stress, and resembled the results found in *Convolvulaceae*, *Medicago truncatula,* and *Jacquemontia pentantha* [62,63,64], but differed from the results reported in *Ipomoea campanulata* [65]. Previous studies have shown that miR398 was crucial for plant stress responses, as it regulates copper-zinc superoxide dismutase (CuZnSOD) levels [66]. Similarly, the present study detected that miR398b targeted UDP-glucosyltransferase (UGT) HRA25 to enable drought stress response. Reduction in the expression level of mtr-miR398a-5p causes increase in the activity of CuZnSODs by achieving the oxidation protection of plants [65]. Exogenous NO silenced the expression of mtr-mir398a-5p, which consequently induced CuZnSODs. Physiological studies have explained that exogenous NO can remove ROS by increasing the activity of SOD [17].

Under drought stress, exogenous NO induced or silenced some conserved miRNAs, which regulate some novel miRNAs. unconservative_c109154.graph_c1_8010 and unconservative_c121624.graph_c0_25405 were down-regulated by exogenous NO, and participated in the metabolic and oxidation-reduction process targeting peroxisome and L-ascorbate oxidase. The novel miRNAs results contribute to further understanding of the regulatory mechanism of exogenous NO under drought stress, as well as supplementing the existing miRNAs. GO enrichment and KEGG pathway analysis were conducted on the target genes of differentially expressed miRNA to further clarify their functions and pathways involved in alfalfa’s response to drought stress. Water shortages along with hydrogen peroxide and respiratory burst were involved in defense response and carbohydrate transport. These were significantly found and have confirmed to be associated with abiotic stresses in many plants [67]. Existing reports confirm that abscisic acid is involved in the metabolism of hydrogen peroxide (H_2_O_2_) and calcium-dependent protein kinase (AtCPK1) to improve the drought resistance of plants [67,68]. Drought responses of carbohydrate metabolism may be important for plant acclimation to stress—this was later verified in *Oryza sativa* [53].

This study reported that carbohydrate transport was significantly enriched in alfalfa under drought stress and that their target genes were positively regulated. Accelerated carbohydrate transport can promote the distribution of carbohydrates in cells and tissues to aid in improving the drought resistance of alfalfa. It is worth mentioning that the metabolic pathways of starch and sucrose (ko00500) have also been enriched in this study. Other studies have shown that inhibition of starch synthesis can directly lead to an increase in soluble sugars of sucrose and glucose [69,70]. The rapid increase of soluble sugars, such as sucrose, in plant cells can form a vitreous state to prevent cell collapse, limiting the mixture of macromolecules in a stable and static state of cells [71]. The present results explain that the exogenous nitric oxide can induce more biological processes to enable participation in the drought resistance of alfalfa. It was further confirmed that exogenous nitric oxide induced more biological processes for active participation in the drought resistance of alfalfa. Among them, mtr-miR5752a was particularly induced and up-regulated by exogenous NO under drought stress, and its target gene could regulate negatively by starch biosynthetic (GO:0019252) and amylopectin biosynthetic process (GO:0010021) to cope with drought stress. This provides important evidence and ideas for the study of the effects of exogenous NO on alfalfa drought resistance.

## 5. Conclusions

A total of 90 discovered miRNAs and 177 unknown miRNAs were found in this experiment. The 14 discovered miRNAs and 18 uncharted miRNAs in the drought response of alfalfa further supplemented the drought-responsive miRNAs. The results provide a lucid understanding of the drought response mechanism of alfalfa and further explain the theoretical basis and technical support for molecular breeding and genetic improvement of alfalfa. In total, 24 known miRNAs and 31 novel miRNAs responded to exogenous nitric oxide under drought stress. Similar to the target genes, the researchers indicated the function of multiple miRNAs by GO enrichment and KEGG analysis. Thus far, this research is the first to investigate the mechanism of exogenous NO regulating plant drought-resistant miRNAs under drought stress, and also provides a better understanding for the action of the exogenous NO mechanism and its application. Simultaneously, the study has reported important scientific value and practical significance for revealing the method of exogenous nitric oxide improving plant drought stress resistance.

## Figures and Tables

**Figure 1 genes-11-00030-f001:**
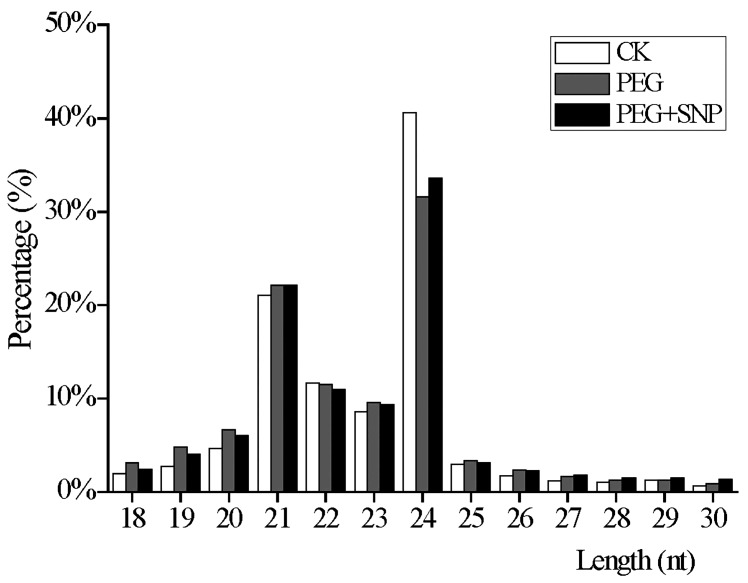
The length distribution of sRNAs in the libraries of CK, PEG, and PEG + SNP in alfalfa.

**Figure 2 genes-11-00030-f002:**
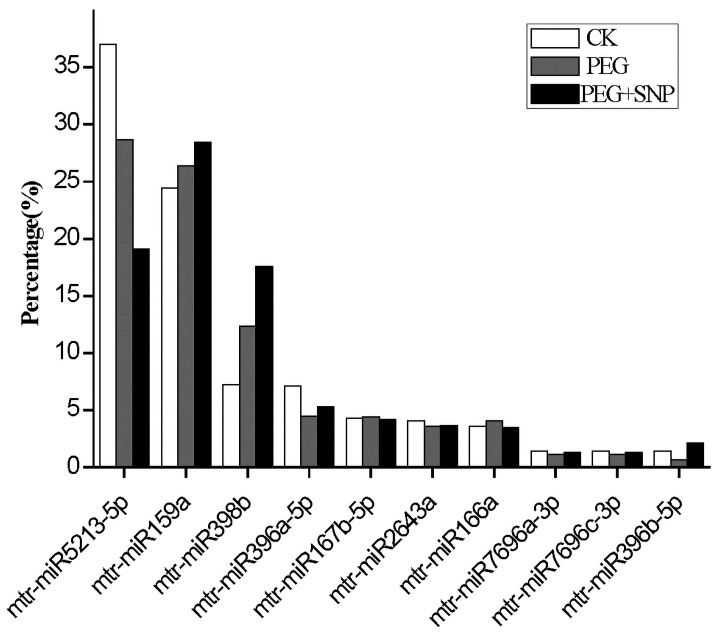
Abundance of the top 10 highly-expressed conserved miRNAs in the seven alfalfa libraries.

**Figure 3 genes-11-00030-f003:**
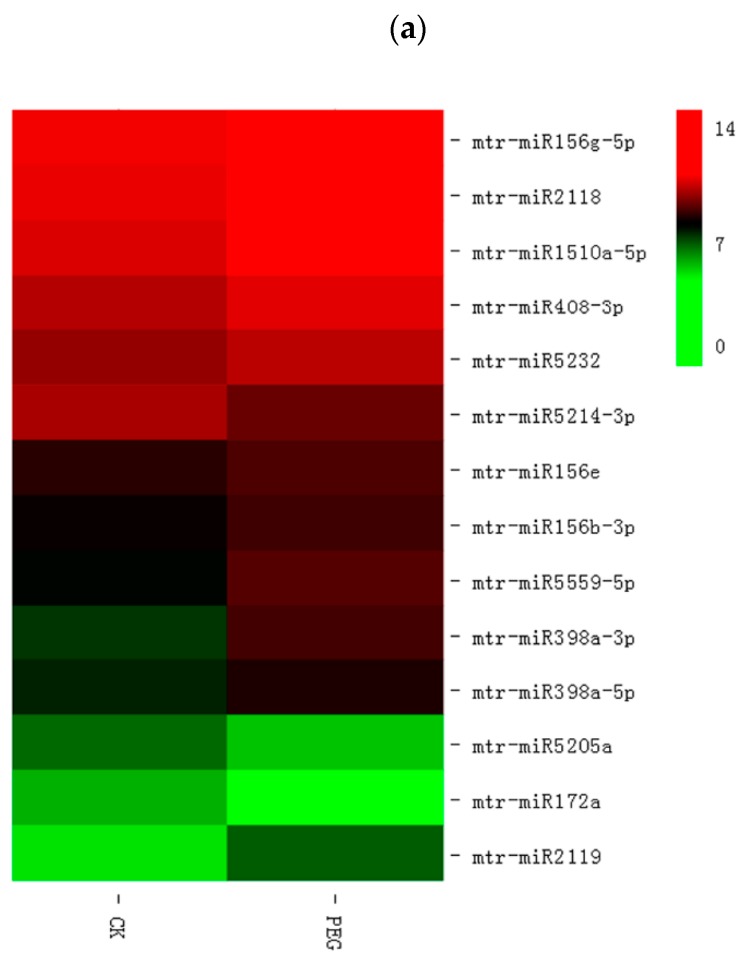
Differentially-expressed conserved miRNAs upon different treatment in alfalfa. (**a**) Conserved miRNAs from CK and PEG samples. (**b**) Conserved miRNAs from CK and PEG + SNP samples. The upregulated miRNAs are showed in red, whereas downregulated miRNAs are shown in green.

**Figure 4 genes-11-00030-f004:**
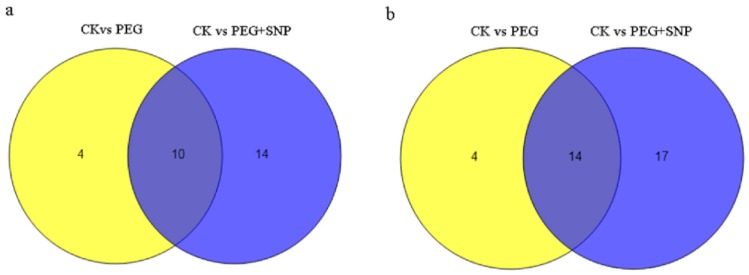
Diagrams of alfalfa miRNAs identified in the three samples: (**a**) conserved miRNAs, (**b**) novel miRNAs.

**Figure 5 genes-11-00030-f005:**
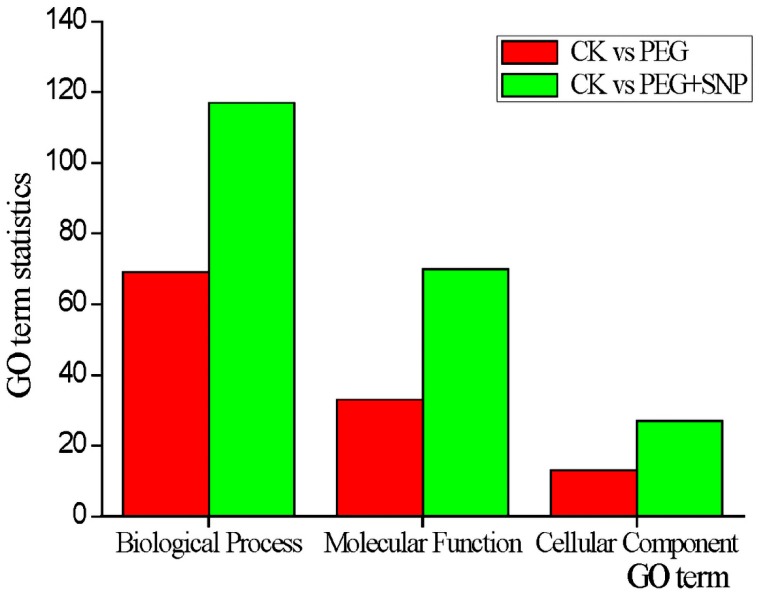
The number of gene ontology (GO) terms in different comparisons.

**Figure 6 genes-11-00030-f006:**
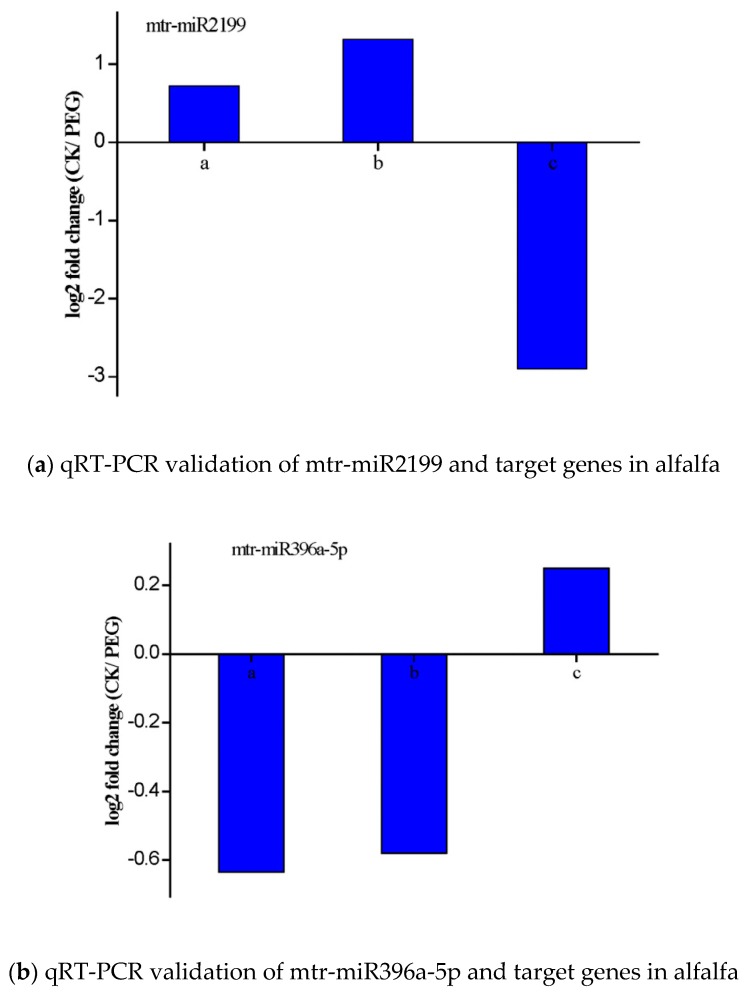
qRT-PCR validation of nine miRNAs and target genes in alfalfa. (a in the x-coordinate) miRNA expression patterns from deep sequencing, (b in the x-coordinate) miRNA expression patterns from qRT-PCR, (c in the x-coordinate) target genes expression patterns.

**Table 1 genes-11-00030-t001:** Alfalfa small RNA (sRNA) sequencing datasets. Statistics of sRNA sequences for CK (normal water), PEG (polyethylene glycol-6000) drought stress, and PEG + SNP (sodium nitroprusside; drought stress and SNP treatment) libraries from *Medicago sativa* L. ‘‘Sandeli’’. Percentage refers to the net read code, which is equal to the ratio of each category to raw reads. rRNA: filtering ribosomal RNA, snoRNA: nucleolar small RNA, tRNA: transport RNA.

Category	CK		PEG		PEG + SNP	
	Count	Percentage	Count	Percentage	Count	Percentage
Raw reads	26,849,326	100.00%	24,252,135	100.00%	20,514,941	100.00%
Clean reads	23,528,741	87.63%	18,046,004	74.41%	15,076,692	73.49%
Mapped to genome	10,366,134	38.61%	7,651,367	31.55%	6,487,853	31.63%
miRNAs	1,267,908	4.72%	1,180,088	4.87%	871,935	4.25%
rRNA	2,931,163	10.92%	3,527,200	14.54%	1,974,535	9.62%
snoRNA	3746	0.01%	7001	0.03%	10,497	0.05%
tRNA	234,634	0.87%	333,859	1.38%	417,378	2.03%
Without annotation	20,273,991	75.51%	14,113,193	58.19%	12,602,624	61.43%

**Table 2 genes-11-00030-t002:** Ten novel miRNAs identified from *Medicago sativa* L. transcriptional databases.

Name	Sequence	Score_Total	Expressional Reads	G + C Contents	Hairpin Energy	Length
c104339.graph_c0_3174	UUUCAUUCCAUAUGUCAUCUAG	17,723.5	14,392	31.82%	−53.6	22
c111080.graph_c1_10087	GGCGGAUGUAGCCAAGUGGA	1,721,964.4	730,326	60.00%	−53.2	20
c114875.graph_c0_14536	UUCAUUUCUAAAAUAGGCAUUG	9543.7	7780	27.27%	−59.2	22
c116131.graph_c0_16318	CGGGAUCGGAGAUUAGAGAAU	146,625.1	106,408	47.62%	−73.0	21
c86020.graph_c0_42677	UACGUCUCUGUCUUUCGGGUUG	16,890.1	13,750	50.00%	−74.6	22
c99918.graph_c0_48282	UUAAUCAAGGAAAUCACAGUC	13,503.9	13,046	33.33%	−55.0	21
c102504.graph_c0_1594	AUGGUUCUUGUUCAGUAGAGU	30,812.8	6815	38.10%	−78.4	21
c91116.graph_c0_44142	UUGUGGAACAUAGAAGCACGUG	9771.1	2317	45.45%	−63.6	22
c121624.graph_c0_25405	UUGAUUCUCAUCACAACUUGG	14,278.7	7926	38.10%	−66.2	21
c111105.graph_c0_10123	UUUGGCAUUCUGUCCACCUCC	8328.7	7532	52.38%	−55.6	21

**Table 3 genes-11-00030-t003:** Ten differentially expressed conserved miRNAs that can specifically induced or silenced by exogenous nitric oxide, including the miRNA name, sequences, target gene or protein, and description of function.

miRNAs	Sequences	Target Gene or Protein	Description of Function	Reference
mtr-miR156a	tgacagaagagagagagcaca	SQUAMOSA promoter-binding-like protein (SPL) genes.	Anthocyanin biosynthesis; vegetative phase transition.	[30,36,37]
mtr-miR399a	tgccaaaggagatttgcccag	Phosphate transporter; high affinity inorganic phosphate transporter.	Pi uptake.	[38,39]
mtr-miR399c	tgccaaaggagatttg			
mtr-miR399q	ccctgtgccaaaggagagctgctctt			
mtr-miR5213-5p	tacgtgtgtcttcacctctgaa	Disease resistance protein.	Response to salt/alkali stress.	[40,41]
mtr-miR5752a	cattgtttggtttagtacaaa	Starch synthase; amino acid binding; metal ion binding.	Starch, galactolipid, amylopectin biosynthetic; response to hypoxia; regulation of ethylene-activated; metabolic process; metal ion transport.	[42]
mtr-miR7696a-5p	tcaagttctcataattcaaaa	Chitin binding; protein kinase.	Innate immune response; protein phosphorylation; cell wall macromolecule catabolic.	[43]
mtr-miR398a-5p	ggagtgacactgagaacacaag	Cu/Zn-superoxide dismutase copper chaperone.	Defense against reactive oxygen toxicity.	[44]
mtr-miR5232	tacatgtcgctctcacctgaa	Glycoside hydrolase; type IIB calcium ATPase protein kinase.	Participate in metabolism.	[41]
mtr-miR5559-5p	tacttggtgaattgttggatc	inorganic diphosphatase magnesium ion binding.	Response to cadmium ion; phosphate-containing compound metabolic.	[29]

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
