# Peer review of "Identification of Exogenous Nitric Oxide-Responsive miRNAs from Alfalfa (Medicago sativa L.) under Drought Stress by High-Throughput Sequencing"

_genes, 2019, doi:10.3390/genes11010030_

Round 1

Reviewer 1 Report

The manuscript studying differential responses of alfalfa plant to NO using high throughput sequencing of miRNA. The topic is interesting, and the subject matter is of interest to the journal readership, however, I felt confused particularly while reading results section and there is a lot of redundancy in the way it's written. I think it needs re-organization and re-writing to be clearer specifically when talking about numbers. Also, I think authors used words to create the confusion as "differentially conserved miRNA"

As there were some key problems in the manuscript. I recommend doing extensive revision and re-evaluation of the manuscript for the following reasons:

Generally, the English language is good even contains few errors in spelling, abbreviation but the parts of the abstract and the results need to be re-written and re-presented as it makes a great confusion. A clear, new and full hypothesis is not presented and, without a robust hypothesis. Besides, hypotheses are important to define treatments, evaluate the methods used and guide the discussion section. Materials and methods missing full information and citation to software, databases and online tools used in the data analysis and have an inadequate description in section 2.6. The statistical significance test, the method to compare data or standard errors was not mentioned in all figures. Legend of figures should be more descriptive so that the reader does not need to go back to the text to interpret it. All scientific names are not in italics in the references section. Files of the supplementary materials were not provided.

Abstract:

lines 17-20, numbers are not clear, the authors said that they have identified 177 novel miRNA. but the count is not right, 32 in PEG samples and 55 in PEG+SNP equal 87 so 90 miRNA are missing and if they are the conserved miRNA, authors should mention them after mentioning the number 177 and anyway this paragraph should be rewritten. The authors did not unify abbreviations as in miRNA, mirnas or microRNA in line 207.

Introduction: Some abbreviations did not identify as in lines 33 and 55, .. or identified with two different descriptions as SNP in lines 15 and 43 and after that. Citation no. 7: this ref about using NO and you talked about it in the next paragraph so you need to replace this ref with another general example or with more than example used other materials as polyamines, boron,...etc.

Materials and Methods: In lines 97-98, “Plants in the CK and PEG samples were watered with 5 ml of water daily at the same time” why? if you mean that you provide the same volume of spraying provided for PEG+SNP group, it was better to spray CK and PEG with 5 ml of water. Citation is not provided for databases or software as in lines 115, 118, 134 Authors did not describe the qRT-PCR fully and a lot of details are missed as, the volume of the template that used in synthesizing cDNA, information about the real-time PCR manager or the thermocycler, the dilution factor if dilution was applied to the cDNA samples, the number of biological and technical replicates? Did you do the standard curve for the examined q-PCR or not?

Results: Percentages in Table 1 are confusing, please clarify how did you calculate the % of each category of RNA, should they equal to % of mapped reads to the genome or equal to the clean reads? you have to mention table 2 first or change the arrangement of tables and put table 3 then table 2 and renumbering them. Mentioning to S3 was missed and all supplementary materials were not provided. The scientific name in line 204 is not in italics.

Discussion: Lines 380-381: “Our results once again demonstrated that the regulatory mechanism of miRNAs in the same species varies with the degree of drought, plant growth stage and genotypes” how?? did you use more than the growth stage? or more than genotype? Supplementary materials were not fully described at the end of the manuscript. References contain a lot of scientific names which all not in italics.

Author Response

Response to Reviewer Comments

Point 1: The manuscript studying differential responses of alfalfa plant to NO using high throughput sequencing of miRNA. The topic is interesting, and the subject matter is of interest to the journal readership, however, I felt confused particularly while reading results section and there is a lot of redundancy in the way it's written. I think it needs re-organization and re-writing to be clearer specifically when talking about numbers. Also, I think authors used words to create the confusion as "differentially conserved miRNA"

Response 1: In order to make the result more clear, we have changed some of the writing in the result according to your opinion, especially in the aspect of numbers. Please see the manuscript for details.Differentially-expressed conserved miRNAs in the manuscript we want to refer to differentially expressed miRNAs belonging to known miRNAs.

Point 2: As there were some key problems in the manuscript. I recommend doing extensive revision and re-evaluation of the manuscript for the following reasons.

Response 2: We have completely revised and rewritten the manuscript. Please see the manuscript for details.

Point 3: Materials and methods missing full information and citation to software, databases and online tools used in the data analysis and have an inadequate description in section 2.6.

Response 3:We have added the missing information, software, database and online tool references in the corresponding position of the manuscript.

Point 4: The statistical significance test, the method to compare data or standard errors was not mentioned in all figures.

Response 4:We used semi-quantitative PCR for verification, and what we want to show in the results is that the expression pattern of miRNAs and their target genes is opposite. If you think that standard error is still needed here, we will label it in the figure.

Point 5: Legend of figures should be more descriptive so that the reader does not need to go back to the text to interpret it. All scientific names are not in italics in the references section. Files of the supplementary materials were not provided.

Response 5:We have modified part of the legend to add a description of the figure. See manuscript for details.

 Point 6: All scientific names are not in italics in the references section. Files of the supplementary materials were not provided.

Response 6:We have changed all scientific names in the manuscript to italics and refined the required supplemental materials.

 Point 7: lines 17-20, numbers are not clear, the authors said that they have identified 177 novel miRNA. but the count is not right, 32 in PEG samples and 55 in PEG+SNP equal 87 so 90 miRNA are missing and if they are the conserved miRNA, authors should mention them after mentioning the number 177 and anyway this paragraph should be rewritten. The authors did not unify abbreviations as in miRNA, mirnas or microRNA in line 207.

Response 7: We have rewritten the abstract. The differentially expressed mirnas identified in PEG and PEG+SNP samples were both known and new. The 32 differentially expressed mirnas identified in the PEG sample contained 14 known mirnas and 18 new ones. The 55 differentially expressed mirnas identified in the PEG+SNP samples included 24 known mirnas and 31 new ones. Not all the differentially expressed mirnas came from 177 new mirnas. And we have replaced the microRNA in the manuscript with miRNA.

Point 8: Introduction: Some abbreviations did not identify as in lines 33 and 55, .. or identified with two different descriptions as SNP in lines 15 and 43 and after that. Citation no. 7: this ref about using NO and you talked about it in the next paragraph so you need to replace this ref with another general example or with more than example used other materials as polyamines, boron,...etc.

Response 8:We have commented on the abbreviations in lines 33 and 55 and deleted the SNP brackets in line 43 to unify the description. In addition, we used other references instead of [7].

Point 9: Materials and Methods: In lines 97-98, “Plants in the CK and PEG samples were watered with 5 ml of water daily at the same time” why? if you mean that you provide the same volume of spraying provided for PEG+SNP group, it was better to spray CK and PEG with 5 ml of water.

Response 9:We watered plants in CK and PEG samples with 5 ml of water per day in order to provide the same spray amount as the PEG+SNP group.

Point 10: Citation is not provided for databases or software as in lines 115, 118, 134 Authors did not describe the qRT-PCR fully and a lot of details are missed as, the volume of the template that used in synthesizing cDNA, information about the real-time PCR manager or the thermocycler, the dilution factor if dilution was applied to the cDNA samples, the number of biological and technical replicates? Did you do the standard curve for the examined q-PCR or not?

Response 10:All the database and software we use have been added to the corresponding database url or software version. In this paper, we have perfected the process of qrt-pcr, including the volume of template for cDNA synthesis, as well as the parameters related to the thermal cycle of PCR. We performed three technical replications and analyzed the data according to the relative quantitative method, with the standard curve.

Point 11: Results: Percentages in Table 1 are confusing, please clarify how did you calculate the % of each category of RNA, should they equal to % of mapped reads to the genome or equal to the clean reads? you have to mention table 2 first or change the arrangement of tables and put table 3 then table 2 and renumbering them. Mentioning to S3 was missed and all supplementary materials were not provided.

Response 11:We modified the percentage in table 1 to be equal to net read code, which is the ratio of counts to raw reads counts for each category. We adjusted the logical order when writing the results section and brought out table S3 in the corresponding position. We have added complete supplementary materials and described them at the end of the manuscript.

Point 12: Discussion: Lines 380-381: “Our results once again demonstrated that the regulatory mechanism of miRNAs in the same species varies with the degree of drought, plant growth stage and genotypes” how?? did you use more than the growth stage? or more than genotype? Supplementary materials were not fully described at the end of the manuscript. References contain a lot of scientific names which all not in italics.

 Response 12:We have modified the inaccurate description in this part, and added complete supplementary materials and described them at the end of the manuscript. The scientific names in the references have been changed to italics. Please refer to the manuscript for details.

Reviewer 2 Report

Manuscript ID: genes-638122

Title:

Identification of Exogenous Nitric Oxide-Responsive miRNAs from Alfalfa (Medicago sativa L.) under Drought Stress by High-Throughput Sequencing

The idea of identify miRNAs that respond to NO is interesting, however, it is not clear in the discussion, the response or novelty of the study, as well as the importance of doing it in alfalfa. Is neccesary review this part.

A description of the results is observed, but there is no clear the discussion, please check this.

Authors mention some important points of  miRNAs that could have an important function in alfalfa, authors not present a robust discussion with the literature,  there is already information about the role of NO.

I Detect a lot erros in the manuscript. Errors are indicate in the manuscript

Some sentences are incomplete or poorly written.

I detect grammatical errors

Sorry, but I did not find the database in SAR, with the number asigned to data. 

authors indicate that they have 9 libraries and discard 2, however, after depuration, authors only present three values, they do not indicate what happened, if it is an average of the replicas for example.

I consider it is necessary to review the results in order to rewrite the results and discussion.

Several details of the methodology are omitted, for example, information about the depuration process and library analysis, for example:

Line 104:  Agilent 2100 bioanalyzer (details of equipment)

Line 106: The RNA samples ligated with adapters were reverse-transcribed and amplified by PCR to produce sequencing libraries…. What adapters? What kit was used?

Line 111: A 50 nt length of raw reads was sequenced. The raw reads were filtered to remove sequences

112 that contained adapters, insertions, poly (A) tails, and reads smaller than 18nt. Question, How removed the adapters? Several details are omited.

Line 265: Genes must be write using italics letters

An important point (Line 372…) The authors indicate that “miR5213 is an important miRNA, and its expression in even greater than conserved miRNAs”; however, they do not quantify it in real time PCR, not validate its target genes, it would be very worth knowing if really what its possible target gene and its expression in the different treatments. This analysis could be robust this afirmation

Sometimes there are missing references that support the statements that are used

Some phrases are disperse, there is no sequence with the results and discussion. For example:

-Line 294 “Compared with drought stress, exogenous NO particularly regulated carbohydrate metabolism, which included cellulose biosynthetic process (GO:0030244), starch biosynthetic process (GO:0019252), galactolipid biosynthetic process (GO:0019375), amylopectin biosynthetic process (GO:0010021) and carbohydrate transmembrane transport (GO:0034219)….Which is interesting, however there is no discussion of this part that could be relevant.

-Authors indicate that “mtr-miR5213-5p is of great significance for alfalfa to adapted abiotic stresses”. However no exist a complete discussion of this miRNA,

-Line 377- The possible reason is that novel miRNAs can only be induced to express under certain conditions, and we think that its expression mechanism can still be further explored and discovered”..

- line 440 “Some transcription factors or functional enzymes act as target genes and play an important role in stress response”. This are general ideas, but not are ligated to results.

Some statements or comparisons are not referenced. Is necesary include appropiate references For example:

“Defense response, cope with water shortages, cope with hydrogen peroxide, respiratory burst involved in defense response and carbohydrate transport were significantly found, these have been confirmed to be associated with abiotic stresses in many plants.” Many?? What studies showed this???

Author Response

Response to Reviewer Comments

Point 1: The idea of identify miRNAs that respond to NO is interesting, however, it is not clear in the discussion, the response or novelty of the study, as well as the importance of doing it in alfalfa. Is neccesary review this part. 

Response 1: We gave a brief explanation at the end of the discussion and gave the meaning of innovation and research in the conclusion.

Point 2: A description of the results is observed, but there is no clear the discussion, please check this.

Response 2: We have revised and improved the discussion section. Please see the manuscript discussion section for details.

Point 3: Authors mention some important points of  miRNAs that could have an important function in alfalfa, authors not present a robust discussion with the literature,  there is already information about the role of NO.

Response 3: We mentioned in the manuscript that the research confirms that NO has a positive effect on improving plant stress tolerance, and explains why we use exogenous NO treatment, but our focus is to identify miRNAs that respond to NO. If you still think that there should be a discussion of NO here, we will add it.

Point 4: I Detect a lot erros in the manuscript. Errors are indicate in the manuscript. Some sentences are incomplete or poorly written. I detect grammatical errors.

Response 4: We improved some of the statements and corrected the syntax.

Point 5: Sorry, but I did not find the database in SAR, with the number asigned to data.

Response 5: Since our data has not been released, we cannot directly check it in NCBI, but the submission can be reviewed in submission portal under the submitters account on the "Manage Data" page:https://dataview.ncbi.nlm.nih.gov/object/PRJNA551564.

Point 6: authors indicate that they have 9 libraries and discard 2, however, after depuration, authors only present three values, they do not indicate what happened, if it is an average of the replicas for example.

Response 6: In order to ensure the accuracy of subsequent analysis, we removed the two libraries with poor correlation and finally took the average value of the three treatments.

Point 7: I consider it is necessary to review the results in order to rewrite the results and discussion.

Response 7: We have reviewed and rewritten the results and discussions. Please refer to the manuscript for details.

Point 8: Several details of the methodology are omitted, for example, information about the depuration process and library analysis, for example: Line 104:  Agilent 2100 bioanalyzer (details of equipment), Line 106: The RNA samples ligated with adapters were reverse-transcribed and amplified by PCR to produce sequencing libraries…. What adapters? What kit was used? Line 111: A 50 nt length of raw reads was sequenced. The raw reads were filtered to remove sequences. 112 lines that contained adapters, insertions, poly (A) tails, and reads smaller than 18nt. Question, How removed the adapters? Several details are omited.

Response 8: We have revised the methods section and added some details. The details of the bioanalyzer device are added in line 104, and the adapter in line 106 is the sequence that precedes the sequence sequence. Each sample has an adapter to identify and distinguish the relationship between the sample and the sample. The connection adapter USES the matching buffer and Enzyme Mix. The adapter in line 112 is deleted using the USER enzyme.

Point 9: Line 265: Genes must be write using italics letters.

Response 9: We have italicized all the scientific names in the manuscript.

Point 10: An important point (Line 372…) The authors indicate that “miR5213 is an important miRNA, and its expression in even greater than conserved miRNAs”; however, they do not quantify it in real time PCR, not validate its target genes, it would be very worth knowing if really what its possible target gene and its expression in the different treatments. This analysis could be robust this affirmation.

Response 10: Six known mirnas, three new mirnas and their target genes were selected for real-time quantitative PCR validation, but miR5213 and its target genes were not included in the selected list. In the discussion, we mentioned that miR5213 is an important miRNA, because in reading other literatures, we found that miR5213 also has a positive effect on alfalfa's saline-alkali stress resistance, but this is not an accurate statement, and we have modified this part of the article.

Point 11: Sometimes there are missing references that support the statements that are used.

Response 11: We added references to some of the quotes.

Point 12: Some phrases are disperse, there is no sequence with the results and discussion. For example:Line 294 “Compared with drought stress, exogenous NO particularly regulated carbohydrate metabolism, which included cellulose biosynthetic process (GO:0030244), starch biosynthetic process (GO:0019252), galactolipid biosynthetic process (GO:0019375), amylopectin biosynthetic process (GO:0010021) and carbohydrate transmembrane transport (GO:0034219)….Which is interesting, however there is no discussion of this part that could be relevant.

Response 12: We have included a discussion of this section in the manuscript, please see the manuscript for details.

Point 13: Authors indicate that “mtr-miR5213-5p is of great significance for alfalfa to adapted abiotic stresses”. However no exist a complete discussion of this miRNA.

Response 13: We modified the description of mtr - miR5213-5p.

Point 14: Line 377- The possible reason is that novel miRNAs can only be induced to express under certain conditions, and we think that its expression mechanism can still be further explored and discovered”..

Response 14: We believe that conserved miRNA regulates many target genes in important metabolic processes in plants, and the expression of new miRNA may be induced by environmental factors or due to tissue specificity.

Point 15: line 440 “Some transcription factors or functional enzymes act as target genes and play an important role in stress response”. This are general ideas, but not are ligated to results.

Response 15: We have deleted the sentence and modified the part. Please refer to the manuscript for details.

Point 16: Some statements or comparisons are not referenced. Is necesary include appropiate references For example:“Defense response, cope with water shortages, cope with hydrogen peroxide, respiratory burst involved in defense response and carbohydrate transport were significantly found, these have been confirmed to be associated with abiotic stresses in many plants.” Many?? What studies showed this???

Response 16: We have added the corresponding references to the manuscript. Please refer to the manuscript for details.

Round 2

Reviewer 1 Report

Dear Editor,

The manuscript has been significantly improved and now warrants publication in Genes

Author Response

Dear reviewer:

      Thank you very much for your comments on the article all the time, and I am honored that our manuscript can be approved by you. Since receiving your first reply, we have carefully read each of your suggestions and made corresponding modifications. Your comments and responses are essential to making significant improvements to the manuscript. Every reply from you will add motivation and information to my work, and we look forward to the day when the article will be published. Thank you again for your tolerance and comments. I wish you a smooth work and a happy life.

Reviewer 2 Report

Manuscript ID: genes-638122. Identification of Exogenous Nitric Oxide-Responsive miRNAs from Alfalfa (Medicago sativa L.) under Drought Stress by High-Throughput Sequencing

After reviewing the manuscript, I can mention the following:

The manuscript was improved considerably, however, there are still many punctuation nomenclature mistakes, some are highlight in the manuscript. I consider it very important to carefully review the writing and grammar and it is highly recommended to send with a professional in English

The contribution is important, however, it is necessary to present the information properly, and improve the discussion

It is necessary to correct the nomenclature errors, for example, using microRNA, mirna, miR miR

Line 387: Authors suggest interesting ideas, for example:

“Participation of more cellular components will help alfalfa improve the efficiency of drought response ..”

However, it is not clear how the results of the identified miRNAs contribute to these ideas

It is necessary to restructure sentences and use the appropriate punctuation, to be able to read the information clearly, since the structure that authors present it is confusing.

For example:

Line 397-401:

“It must be noted that when PEG and PEG+SNP samples were specifically enriched in 2(RNA transport and endocytosis) and 8(Ascorbate metabolism, tyrosine metabolism, starch metabolism, sucrose metabolism, amino acid synthesis, ribosome and protein transport) pathways respectively, 4(Oxidative phosphorylation, phenylpropanoid metabolism, biosynthesis of ubiquinone and other terpenoids, and alpha-linolenic acid metabolism) pathways were co-owned by them”

I suggest reducing the graphic size in figure 6. These use too much space

Author Response

Point1:The manuscript was improved considerably, however, there are still many punctuation nomenclature mistakes, some are highlight in the manuscript. I consider it very important to carefully review the writing and grammar and it is highly recommended to send with a professional in English.

Reponse1:We have carefully reviewed  the use of grammar and punctuation of the full text with the help of English professionals. Please see the revised manuscript.

Point2:The contribution is important, however, it is necessary to present the information properly, and improve the discussion.

Reponse2:We have partially rearranged and rewritten the results and discussion sections.Please see the revised manuscript.

Point3:It is necessary to correct the nomenclature errors, for example, using microRNA, mirna, miR miR.

Reponse3:We have unified the writing of miRNA in this paper.Please see the revised manuscript.

Point4:Line 387: Authors suggest interesting ideas, for example:

“Participation of more cellular components will help alfalfa improve the efficiency of drought response ..”However, it is not clear how the results of the identified miRNAs contribute to these ideas.

Reponse4:It can be seen from our results that exogenous NO can induce more miRNAs and corresponding target genes. A total of 114 target genes were annotated in the PEG+SNP sample, 55 more than in the PEG sample. In PEG+SNP samples, these target genes obtained 214 enrichment categories through KEEG analysis and GO enrichment, including 27 cell components, 14 more than that in PEG samples. These cellular components are related to plant stress resistance, so we believe that more cellular components can help alfalfa improve drought resistance. It was also proved that NO had positive effect on drought resistance of alfalfa.

Point5:

It is necessary to restructure sentences and use the appropriate punctuation, to be able to read the information clearly, since the structure that authors present it is confusing.For example:Line 397-401:

“It must be noted that when PEG and PEG+SNP samples were specifically enriched in 2(RNA transport and endocytosis) and 8(Ascorbate metabolism, tyrosine metabolism, starch metabolism, sucrose metabolism, amino acid synthesis, ribosome and protein transport) pathways respectively, 4(Oxidative phosphorylation, phenylpropanoid metabolism, biosynthesis of ubiquinone and other terpenoids, and alpha-linolenic acid metabolism) pathways were co-owned by them”.

Reponse5:Your opinion is very important, and we find the writing of these sentences really puzzling. Now we've rewritten this part.Please see the revised manuscript.

Point6:I suggest reducing the graphic size in figure 6. These use too much space.

Reponse6:We have reduced the size of the image in figure 6 to reduce the space.Please see the revised manuscript.

Round 3

Reviewer 2 Report

Dear Editor

Thank you again for the invitation, I consider the manuscript was improved, so I think it is satisfactory